# Monitoring of Ammonium and Nitrate Ions in Soil Using Ion-Sensitive Potentiometric Microsensors

**DOI:** 10.3390/s24227143

**Published:** 2024-11-06

**Authors:** Matthieu Joly, Maurane Marlet, David Barreau, Arnaud Jourdan, Céline Durieu, Jérôme Launay, Pierre Temple-Boyer

**Affiliations:** 1CNRS, LAAS, 7 Avenue du Colonel Roche, F-31400 Toulouse, France; matthieu.joly@laas.fr (M.J.); launay@laas.fr (J.L.); 2INSAT, UT3-PS, INPT, University of Toulouse, 118 Route de Narbonne, CEDEX 9, F-31062 Toulouse, France; 3AGRONUTRITION SA, Parc Activestre, 3 Avenue de l’Orchidée, F-31390 Carbonne, France; m.marlet@agro-nutrition.fr (M.M.); c.durieu@agro-nutrition.fr (C.D.); 4SIREA SA, 1 Rue Jean Perrin, F-81100 Castres, France; d.barreau@sirea.fr (D.B.); arnaud.jourdan@caelimp.fr (A.J.)

**Keywords:** ion-sensitive field-effect transistor, ISFET, potentiometric sensor, ion-sensitive layers, ammonium NH_4_^+^ ion, nitrate NO_3_^−^ ion, soil analysis

## Abstract

Focusing on the ChemFET (chemical field-effect transistor) technology, the development of a multi-microsensor platform for soil analysis is described in this work. Thus, different FET-based microdevices (i.e., pH-ChemFET pNH_4_-ISFET and pNO_3_-ISFET sensors) were realized with the aim of monitoring nitrogen-based ionic species in soil, evidencing quasi-Nernstian detection properties (>50 mV/decade) in appropriate concentration ranges for agricultural applications. Using a specific test bench adapted to important earth samples (mass: ~50 kg), first experiments were done in a lab, mimicking rainy periods as well as nitrogen-based fertilizer inputs. By monitoring pH, pNH_4,_ and pNO_3_ in an acidic (pH ≈ 4.7) clay-silt soil matrix, different processes associated to the nitrogen cycle were characterized over a fortnight, demonstrating comprehensive results for ammonium nitrate NH_4_NO_3_ inputs at different concentrations, water additions, nitrification phenomena, and ammonium NH_4_^+^ ion trapping. Even if the ChemFET-based measurement system should be improved according to the soil(electrolyte)/sensor contact, such realizations and results show the ChemFET technology potentials for long-term analysis in soil, paving the way for future “in situ” approaches in the frame of modern farming.

## 1. Introduction

The nitrogen cycle refers to all the processes involved in the transformation of the various forms of nitrogen (dinitrogen N_2_, organic nitrogen generated by vegetal and animal species, mineral nitrogen in ionic forms, etc.) in the atmosphere as well as the terrestrial and aqueous ecosystems [1]. It plays an essential role in the frame of plant growth and development, especially for the synthesis of amino-acids, proteins, enzymes, and chlorophyll-related molecules [2]. Nevertheless, even if dinitrogen N_2_ is abundant in the atmosphere (volume ratio: 78%), its fixation by plants, mainly limited to the nitrogen ionic forms (i.e., ammonium NH_4_^+^, nitrate NO_3_^−^ and nitrite NO_2_^−^ ions), is complex and bio-energy consuming [3]. As a result, in order to improve the agricultural production yields, nitrogen-based fertilizations, with ammonium nitrate NH_4_NO_3_ salt and urea CO(NH_2_)_2_ organic compound, among others, were thoroughly developed and used [4].

Today, a question arises concerning the real effectiveness of such nitrogen fertilization practices. For example, in the frame of cereal cultures, less than half of the nitrogen supply is actually used by crops [5]. In other words, more than half of the nitrogen fertilization is lost in soil depths through leaching, denitrification, volatilization, or consumption by micro-organisms. The environmental consequences of such losses are numerous [6,7,8,9,10]. Firstly, the nitrates leaching from the soil lead to the pollution of groundwaters, thereby endangering human health [8]. Secondly, surface run-off, erosion, and infiltration of nitrate-rich sources from agricultural fields are responsible for the eutrophication of fresh and marine waters, consequently for the proliferation of certain types of fast-growing plants and algae, and finally for a reduction of biodiversity [9]. Thirdly, emission of nitrogen oxides NO_x_ by denitrification as well as ammonia NH_3_ volatilization contribute respectively to global warming and acid rain production [10]. Since such phenomena were definitively incompatible with the development of sustainable farming adopted in many industrial countries, different regulation policies were applied, in order to prevent such losses and cope with their impacts on the environment, life, and human health. Thus, by defining spreading prohibition periods, and by reducing the agricultural nitrogen inputs, promising results were obtained in nitrate-vulnerable zones, for example, in order to limit acidification as well as eutrophication of European ecosystems [11]. Nevertheless, they also showed the need for monitoring nitrogen fertilization processes at the field level.

In this context associated to precision agriculture, analytical methods were no longer usable, since they are expensive, manpower/time-consuming, and limited in spatial resolution. From another point of view, technologies derived from microelectronics allowed the development of miniaturized, portable, autonomous, real-time, and cheap microdevices. Thus, by applying microtechnologies to agriculture, physical/chemical microsensors and analysis microsystems were successfully developed [12,13]. While dealing with the soil-sampling bottleneck associated to possible modifications of samples due to biofouling, transport from the field to the lab and transfer in liquid phase [14], two different approaches, were emphasized. On the one hand, (micro)sensing platforms were embedded on agricultural vehicles for “on-the-go” soil measurement [15,16,17,18,19,20,21]. On the other hand, multi-sensor (micro)systems were buried in fields for the “in-situ” soil analysis [22,23,24,25,26,27,28,29,30]. Nevertheless, in both cases, due to soil mechanical properties, and therefore to the fragility of some standard electrochemical sensors, solid-state devices were given priority, emphasizing the development of silicon-based technologies in order to integrate ion-sensitive electrodes (ISE), as well as chemical field-effect transistors (ChemFET) [16,17,22,23,30,31]. Thus, through the study of numerous ionophore-rich membranes, ion-sensitive field-effect transistor (ISFET) microsensors were developed, aiming to detect ammonium NH_4_^+^ and nitrate NO_3_^−^ ions in the frame of water and soil analyses [16,30,32,33,34,35,36,37,38,39,40,41,42].

This paper deals with the development of a multi-microsensor measurement platform for the “in-situ” soil analysis. Based on a generic pH-ChemFET (pH-sensitive chemical field-effect transistor) technological platform derived from silicon-based microelectronics, it proposes the realization of pNH_4_-ISFET and pNO_3_-ISFET microdevices for the analysis of nitrogen ionic species in soil. It aims to the monitoring of environmental/agricultural processes associated with the nitrogen cycle in the frame of wheat culture.

## 2. Materials and Methods

### 2.1. Microdevice Fabrication

According to a microfabrication process previously studied [30], silicon technologies were used in order to integrate metallic microelectrodes as well as field-effect transistors (FET) sensing microdevices on 6.5 × 5.5 mm^2^ chips (Figure 1). Thus, P-well, N-channel, SiO_2_/Si_3_N_4_-gate, pH-sensitive chemical field-effect transistors (pH-ChemFETs) were fabricated on 4-inch, (100)-oriented, N-type (500 Ω.cm) silicon wafers. Then, for the penultimate technological steps, dedicated to platinum deposition and etching, a specific lithographic mask was used in order to fabricate simultaneously the pH-ChemFET contact pads as well as a conductivity sensor based on platinum microelectrodes (Pt-µE). Finally, a wafer-level passivation was performed using the photosensitive DF-1050 epoxy resin (purchased from EMS company, Hatfield, PA, USA), leaving the pH-ChemFET sensitive zone uncovered and defining the microelectrode active surfaces. The silicon chips so obtained were stuck on a specially-coated printed circuit board using an epoxy-insulating glue (Figure 2). After wire bonding, packaging was finally performed at the system level using a silicone glop-top in order to adapt the final sensor to soil analysis.

### 2.2. Adaptation of FET-Based Sensors to Ion Detection

According to previous works [30,43], pH-ChemFET devices were adapted for ion detection using fluoropolysiloxane-based ion-sensitive membranes (FPSX 730 FS purchased from Dow Corning, Midland, MI, USA). Nonactin and tetradocecylammonium nitrate (TDDAN) ionophores were used, according to specific processes, in order to integrate FPSX-based ion-sensitive layers [30]. All chemical reagents were purchased from Sigma-Aldrich (Saint Louis, MO, USA). Thus, the detection of ammonium NH_4_^+^ and nitrate NO_3_^−^ ions was demonstrated for concentration ranges suitable for soil analysis, and FPSX-based NH_4_^+^-sensitive and NO_3_^−^-sensitive field-effect transistors (respectively called pNH_4_-ISFET and pNO_3_-ISFET hereafter) were realized.

### 2.3. Realization of a Multi-ISFET Platform for Soil Analysis

A specific potentiometric measurement interface, based on a “source-drain follower” electronic circuit [30], was realized in order to be useable with six different ChemFET-based microsensors (Figure 2 and Figure 3). Thus, a continuous multi-measurement was possible, provided the correct application of the same ChemFET-gate voltage to the analyzed sample. In the frame of soil analysis, a WE200 reference electrode (purchased from Silvion limited) was chosen to apply this gate voltage, fixed at zero potential (V_reference_ = V_soil_ = V_G_ = 0 V). Indeed, this device, associated to an Ag/AgCl electrode and containing a 0.5 M sodium chloride NaCl solid electrolyte, was a priori designed for permanent use in soils in the frame of oil exploration. It was specifically characterized, using a XR110 commercial calomel reference electrode (purchased from Radiometer Analytical), in order to check its electrochemical properties. This study applies the use of potassium chloride KCl-based solutions (10^−3^–10^−1^ M) with a background electrolyte of lithium acetate CH_3_COOLi (0.1 M) in order to define its intrinsic sensitivity to chloride Cl- ions, as well as its burying into an acidic (pH ≈ 4.7) clay-silt soil matrix, to analyze its temporal drift.

The final measurement platform was integrated into a solid metallic stake (Figure 3a) in order to simplify the burying procedure, prevent any undesired breakage, and ensure electrical contact with the soil matrix. In the frame of wheat culture, the rooting depth generally does not exceed thirty centimeters. Therefore, we chose to position three different sensors (i.e., pH-ChemFET, pNH_4_-ISFET, and pNO_3_-ISFET) at two different depths, −15 cm and −45 cm (Figure 3), eventually enabling a “dual horizon” analysis of nitrogen flows in soils.

### 2.4. Electrochemical Characterization of ChemFET-Based Sensors and Soil Measurements

Before any soil analysis took place, all the different ChemFET-based sensors were tested accordingly, in order to check their detection properties in the liquid phase [30]. Thus, titration experiments using hydrochloric acid (HCl: 10^−2^ M) and tetra-methyl-ammonium hydroxide (TMAH: 10^−1^ M) were performed with a background electrolyte (CH_3_COOLi 0.1 M) solution in order to fully characterize the pH-ChemFET analytical response, whereas pNH_4_- and pNO_3_- ISFETs were studied in standard ammonium nitrate NH_4_NO_3_ solutions, while increasing the concentration from 10^−8^ M to 10^−2^ M. Apart from the threshold voltage discrepancy associated to their potentiometric transduction (see hereafter), the following lists the sensitivities and associated linear measurement ranges typically obtained [30]:
pH-ChemFET: sensitivity: 52 ± 2 mV/decade for pH ranging from 2 to 12,pNH_4_-ISFET: sensitivity: 56 ± 2 mV/decade in the [10^−5^–10^−2^ M] concentration range,pNO_3_-ISFET: sensitivity: 56 ± 2 mV/decade in the [10^−5^–10^−2^ M] concentration range.

In order to perform free soil analysis (i.e., independent of seasonal constraints associated with intensive field farming), a test bench was designed to characterize the ISFET-based measurement platform while controlling the environmental parameters, such as temperature and soil moisture (Figure 3b). It consisted of a PVC tube (diameter: 30 cm, height: 70 cm) with a volume capacity approaching fifty liters, which was supported by a retention tank to prevent any unexpected water leakages and operate water drainage, thanks to specific holes drilled into the bottom. Finally, the test bench was positioned on an electronic scale to measure soil moisture by weight. Once filled with earth, the entire test bench had a mass of around 70 kg.

It should be noted that the WE200 Silvion reference electrode was first tested. As expected, it was characterized by excellent reference properties in liquid phase, with a constant value of around 15.6 mV, compared to the XR110 calomel reference electrode, and a sensitivity to chloride Cl^−^ ions of around 0.5 mV per concentration decade. Nevertheless, in order to ensure its electrochemical behavior in the studied acidic clay-silt soil (pH ≈ 4.7), a specific long-term measurement was performed (Figure 4). Apart from some measurement instabilities from one day to another (±2 mV), possibly due to the soil matrix, the temporal drift was estimated at around 0.11 mV/day, for a total voltage drift of around 18 mV on a 165-day period in soil. According to these results, in the frame of the experiment protocol duration, measurement errors due to the reference electrode drift were estimated at around ±0.8 mV. Since such measurement instabilities will be tackled by the ISFET measurement procedure (see Section 2.4), this fully validated the use of the WE200 Silvion reference electrode for soil analysis experiments.

Finally, soil analysis was performed. The stake-shaped measurement platform, as well as the WE200 reference electrode, was buried in a vertical position into the test bench using around 50 kg of earth. An acidic clay-silt soil (pH ≈ 4.7) directly taken from a wheat field in the south-west of France was chosen for its water drainage and cationic adsorption properties. After burying, “liquid” mud samples were carefully poured around the measurement stake in order to ensure the best soil-sensor electrical contacts. According to previous results [30], the soil-relative moisture was initially set to more than 60%, in order to ensure stable measurement. Since its moisture field capacity was estimated at around 45% mass, 13.5 L of deionized water were therefore added to the 50 kg clay-silt soil sample. Then, the measurement system was started up, just after checking the whole installation as well as all of the different analysis parameters. It was programmed to wake up every five minutes in order to continuously bias the six ISFET sensors for a one-minute period, apply a “zero potential” to the gate contact thanks to the WE200 reference electrode (V_reference_ = V_soil_ = V_G_ = 0 V), and transmit the different output voltage mean values estimated at the end of this measurement period. This procedure was previously shown to decrease the measurement noise and therefore improve the measurement performances [44].

While mimicking events associated to (i) rain, thanks to the addition of deionized water (DI), and (ii) nitrogen fertilization thanks to the addition of ammonium nitrate NH_4_NO_3_ solutions, soil sample monitoring was performed over a 15-day period (corresponding to more than 4000 measurements for each FET-based sensor), according to the following experimental protocol:
Day 0: start of the experiment;Day 0.82: addition of 2 liters of DI water to trigger the soil-sensor electrical contact;Day 1.83: input of 1 liter of an NH_4_NO_3_ solution (1 g/L or 12.5 × 10^−3^ mole);Day 2.15: addition of 1 liter of DI water;Day 5 ± 0.04: addition of 3 liters of DI water to reach soil water saturation;Day 5.17: input of 1 liter of an NH_4_NO_3_ solution (1 g/L or 12.5 × 10^−3^ mole);Days 6.06 and 7.05: addition of 1 liter of DI water;Day 7.98: input of 1 liter of an NH_4_NO_3_ solution (1 g/L or 12.5 × 10^−3^ mole);Days 11.81 and 13.17: addition of 1 liter of DI water;Day 14.23: input of 1 liter of a tenfold-concentrated NH_4_NO_3_ solution (10 g/L or 125 × 10^−3^ mole) to check the final effectiveness of the ion-sensing procedure;Day 15: end of the experiment.

The whole experiment was performed at an ambient temperature (21 °C) in a controlled atmosphere.

## 3. Results and Discussion

### 3.1. pH-ChemFET Characterization in Soil

Both pH-ChemFET sensors were operational for around 3 h (day: 0.12) after the experiment start (i.e., without any addition of deionized water), using only the soil-relative humidity to ensure the electrochemical contact within the Soil (electrolyte)-Insulator-Semiconductor detection structure [30]. This first result confirmed the correct positioning of the WE200 reference electrode in the soil, as well as the gate bias effectiveness for all of the different FET-based sensors.

For the two different measurement depths (i.e., −15 cm and −45 cm), and considering the whole experiment (duration: 15 days), the pH-ISFET output voltages V_out_ were found in the following range:
pH-ChemFET “−15 cm”: Vout = 432 ± 7 mVpH-ChemFET “−45 cm”: Vout = 411 ± 6 mV

The measurement discrepancy between the two pH-ChemFETs (~20 mV) is related to their potentiometric transduction. Indeed, their threshold voltage depends on numerous uncontrolled parameters related to (i) the technological fabrication process, and (ii) the soil(electrolyte)/insulator potentiometric interface, and therefore to the soil matrix intrinsic properties. This “20 mV” potential shift was found constant during the whole experiment (apart from any occasional measurement instabilities).

Furthermore, as shown from Figure 5, Figure 6, Figure 7 and Figure 8 for the “−15 cm” depth, despite any interferences due to the experimental protocol, both pH-ChemFETs showed a temporal drift lower than 1 mV/day. As a result, they were considered as a measurement reference for both pNH_4_-ISFET and pNO_3_-ISFET sensors.

### 3.2. Monitoring Nitrogen-Related Ionic Species in Soil

Contradictory results were obtained for the two different measurement depths. On the one hand, for the “−45 cm” depth, the pNH_4_-ISFET and pNO_3_-ISFET functioned incorrectly from time to time during the experimentation fortnight: their potentiometric variations were unstable (measurement “accuracy”: ±20 mV), discontinuous over daily periods (according to liquid phase inputs or not), and showing finally incomprehensible jumps of tens of millivolts after restarting. Such measurement instabilities were assumed to be due to some damage of the soil-sensor electrical contact, certainly related to the burying/pouring procedure.

On the other hand, for the “−15 cm” depth, all went well for both ISFET sensors. Following the pH-ChemFET (see Section 3.2), the pNH_4_-ISFET and pNO_3_-ISFET sensors were operational after 20 h (day: 0.83) and 30 h (day: 1.25), respectively. In fact, it seems that the first addition of deionized water (volume: 2 liters; day: ~0.83) was responsible for this start, emphasizing the role of water in the improvement of the soil-sensor electrical contact. Then, after this starting period, all of the sensors gave coherent and stable measurement values for all of the soil-monitoring experiment duration (i.e., from day 0 to day 15). In this context, four different chains of events were more carefully studied (Figure 5, Figure 6, Figure 7 and Figure 8).

However, before discussing experimental results, it should be noted that since the NH_4_^+^ and NO_3_^−^ ions are of opposite ionic valence, the pNH_4_-ISFET and pNO_3_-ISFET are characterized by opposite output voltage variations. As a result, an [NH_4_^+^] concentration increase is related to a potential decrease, whereas an [NO_3_^−^] concentration increase is related to a potential increase. In a similar way, a pH increase is associated to an [H_3_O^+^] concentration decrease, and therefore to a pH-ISFET output voltage increase.

The first studied event is associated to the [1.5–2.5] daily period, involving the first input of ammonium nitrate NH_4_NO_3_ (1 L, 12.5 mmol) on day 1.83, followed by deionized water pouring (1 L) on day 2.14 (Figure 5). After the nitrogen input—and as expected—the pNH_4_-ISFET output voltage decreased while the pNO_3_-ISFET output voltage increased. Indeed, taking respectively into account their cationic and anionic properties, these temporal variations are effectively related to an increase of both [NH_4_^+^] and [NO_3_^−^] concentrations in the soil matrix.

Modelling such variations with an “exp (−t/τ)” mathematical model, it was possible to estimate their time constant τ at roughly one hour, giving some information concerning NH_4_NO_3_ fertilization kinetics in soils (τ_fertilization_ ≈ 1 h). Thus, according to this empirical “decreasing exponential” model, and taking into account the detection sensitivity of both ISFET sensors (~56 mV/decade), the different voltage variations, as well as the corresponding concentration multiplication ratios, were estimated as follows:
pNH_4_-ISFET: voltage variation: −56 mV, [NH_4_^+^] multiplication ratio: ≈10pNO_3_-ISFET: voltage variation: +22 mV, [NO_3_^−^] multiplication ratio: ≈2.5

In order to better understand these results, the following simple dilution model was developed while considering that all ions are chemically available in the studied clay-silt soil matrix:$$\left\{\begin{array}{c} \frac{C_{\mathrm{soil}}V_{\mathrm{soil}}+C_{\mathrm{added}}V_{\mathrm{added}}}{V_{\mathrm{soil}}+V_{\mathrm{added}}}=\gamma C_{\mathrm{soil}}\\ V_{\mathrm{soil}}=\alpha V_{\mathrm{added}} \end{array}\right.\Rightarrow C_{\mathrm{soil}}=\frac{C_{\mathrm{added}}}{\gamma \left(1+\alpha \right)-\alpha }$$
where V_soil_ and V_added_ are the volumes of water initially present in and added to the soil sample (V_soil_ ≈ 15.5 L and V_added_ = 1 L, i.e., a ≈ 15.5), C_soil_ and C_added_ are the concentrations in the soil sample and in the added NH_4_NO_3_ solution (C_added_ = 12.5 mM), and g is the corresponding concentration multiplication ratio.

Thus, it was possible to calculate the different ionic concentrations in the studied soil sample according to the previous equations. The initial [NH_4_^+^] and [NO_3_^−^] concentrations were estimated at ≈0.08 mM and ≈ 0.5 mM, respectively, results which are in agreement with previous concentrations obtained by ionic chromatography analysis on a similar clay-silt soil sample [30].

It should be noted that the deionized water addition operated on day 2.14 was then of little influence, except for small inflections evidenced on the experimental curves. Since these phenomena were associated with a concentration decrease for both NH_4_^+^ and NO_3_^−^ ions, they are quite consistent with some dilution effects, even if greater variations were expected (see hereafter).

Finally, apart from some measurement instabilities, the pH-ChemFET output voltage was characterized by non-significant pH variations, remaining in the [427 mV–431 mV] range during the [1.5–2.5] daily period. However, the “NH_4_NO_3_ fertilization” event (day = 1.83) and the “water addition” event (day = 2.14) were both responsible for a significant 1–2 mV positive variation (i.e., corresponding to a 0.06 ± 0.02 pH increase). For the second case, this is in agreement with some dilution phenomenon in an acidic soil. However, for the first case, such a pH increase is in disagreement with an ammonium nitrate NH_4_/NO_3_ input, while considering the acidic property of the NH_4_^+^/NH_3_ couple (pKa = 9.23) and taking into account the clay-silt soil pH-buffer properties.

The second study concerned the dilution cycle operated on days 4.96, 5.00, and 5.04 in order to approach soil water saturation (i.e., 100% relative humidity), followed by the second input of ammonium nitrate NH_4_NO_3_ (1 L, 12.5 mM) operated on day 5.17 (Figure 6). Water saturation of the clay-silt soil sample was checked visually by detecting water in the retention tank. Furthermore, unlike the previous experiment (see Section 2.4), the three deionized water additions were responsible for a significant concentration decrease, as well as some pH variations. Concerning the nitrogen-based ions, both the total variations as well as the corresponding concentration multiplication ratios (lower than 1, as expected for any dilution procedures) were estimated fully as follows:
pNH_4_-ISFET: voltage variation: +6 mV, [NH_4_^+^] multiplication ratio: ≈0.8pNO_3_-ISFET: voltage variation:−16 mV, [NO_3_^−^] multiplication ratio: ≈0.5

**Figure 6 sensors-24-07143-f006:**
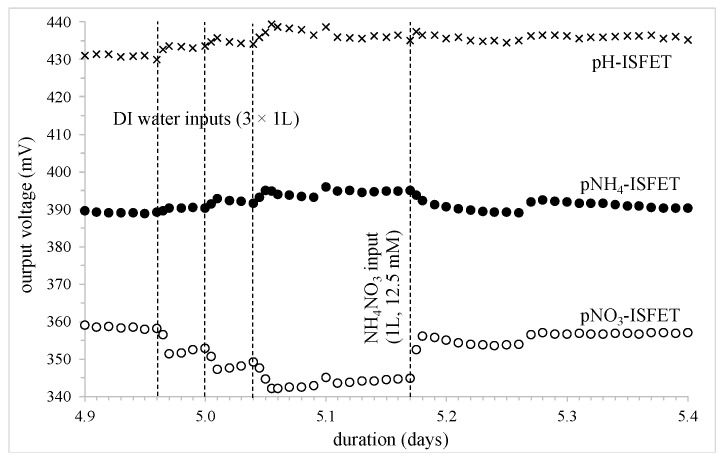
Potentiometric measurements for the different FET-based sensors (depth: −15 cm); the influence of a water saturation cycle followed by an NH_4_NO_3_ input (1 L, 12.5 mmol).

In parallel, the pH-ChemFET output voltage also increased by 10 mV, associated to a pH variation of around 0.2. Since the ammonium NH_4_^+^ ion is a weak acid (pKa = 9.23), the pH and pNH_4_ parameters are necessarily correlated in the frame of the cation adsorption in the negatively-charged clay-humus colloidal complex [45]. As a result, the soil [NH_4_^+^] concentration decrease was hindered by the soil pH increase, explaining the low output voltage variation evidenced for the pNH_4_-ISFET. On the contrary, since there is no buffer effect for nitrate NO_3_^−^ ions, the dilution cycle was responsible for the soil leaching and the [NO_3_^−^] concentration finally being halved.

The second ammonium nitrate NH_4_NO_3_ input operated on day 5.17 was of less influence than the first input, which is characterized as follows:
pNH_4_-ISFET: voltage variation: −6 mV, [NH_4_^+^] multiplication ratio: ≈1.3pNO_3_-ISFET: voltage variation: +12 mV, [NO_3_^−^] multiplication ratio: ≈1.65

For the nitrate NO_3_^−^ ion, according to our experimental analysis, the lower increase ratio should be related to a higher [NO_3_^−^] initial concentration in the soil matrix, and indirectly related to the first ammonium nitrate NH_4_NO_3_ input that occurred on day 1.83 (see before). For the ammonium NH_4_^+^ ion, the dilution decrease was balanced by the nitrogen input, the [NH_4_^+^] concentration unchanged after this chain of events. This result illustrated again the low dynamics of cation variations within the soil clay-humus complex.

Finally, some measurement instabilities can be detected on days 5.1 and 5.27 (Figure 6). Since these phenomena are common to the three different IFET-based sensors, they should be related to uncontrolled potentiometric/conductimetric shifts in the “reference electrode/soil(electrolyte)/ISFET sensor” system.

It should be noted that very similar results were obtained for the following deionized water additions (days 6.06 and 7.05), as well as for the third ammonium nitrate NH_4_NO_3_ input (day: 7.98).

The third study concerned the “dry weather” period that occurred between day 8 and day 11.8 (Figure 7). After the last ammonium nitrate NH_4_NO_3_ input (day: 7.98), it took one day for the pH-ISFET to stabilize at a potential value of 430 ± 1 mV. Then, for almost 3 days, there were no measurement instabilities, demonstrating that the whole system was operating well. Considering a measurement accuracy of around 1 mV (see Section 3.2), it was assumed that the few millivolt variations evidenced for both ISFET sensors were representative of some chemical phenomena.

**Figure 7 sensors-24-07143-f007:**
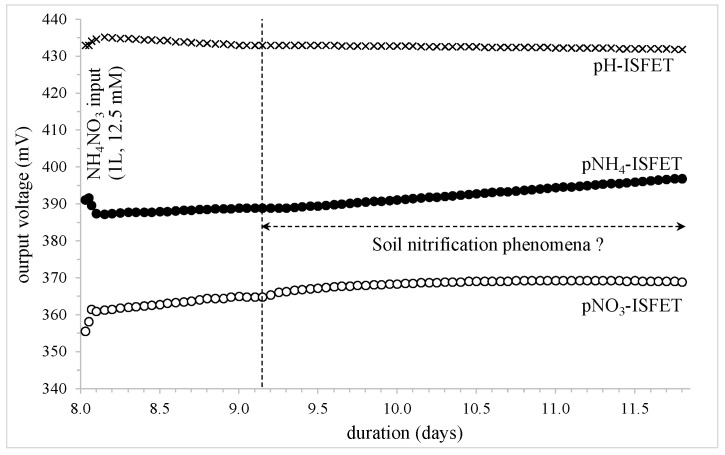
Potentiometric measurements for the different FET-based sensors (depth: −15 cm); the influence of a “dry weather” period.

Indeed, between day 9 and day 11.8, the pNH_4_-ISFET output voltage followed a linear increase (slope: ~3.2 mV/day), whereas the pNO_3_-ISFET output voltage was characterized by “decreasing exponential” kinetics. Thus, once again using a simple mathematical model, the different voltage variations, as well as the corresponding concentration multiplication ratios, were estimated as follows:
pNH_4_-ISFET: voltage variation: +8 mV, [NH_4_^+^] multiplication ratio: ≈0.7pNO_3_-ISFET: voltage variation: +4 mV, [NO_3_^−^] multiplication ratio: ≈1.2

From our knowledge [8], this coupled phenomena should be linked to biochemical processes that are ultimately responsible for the clay-silt soil nitrification due to the oxidation of ammonium ion in nitrate ion: NH_4_^+^ + O_2_ + 7H_2_O -----> NO_3_^−^ + 6H_3_O^+^ + 4e^−^.

According to our experimental results, the NH_4_^+^ ion oxidation rate was estimated at around 0.6 mM/day, and the time constant τ of the nitrification kinetics was found at around 0.5 days, giving some further information concerning this chemical phenomenon in soils (τ_nitrification_ ≈ 10 h).

Finally, it should be mentioned that the nitrification-related effect phenomena were only slightly affected by the following deionized water additions (days 11.81 and 13.17), and was finally evidenced up to the last ammonium nitrate NH_4_NO_3_ input (day 14.23). Furthermore, this nitrification process should be responsible for a pH decrease that was not detected by the pH-ISFET. Again, this should be related to the acidity as well as buffer properties of the studied soil sample.

For the last deionized water additions (days 11.81 and 13.17), the same results were obtained as for previous additions, but the last event, occurring on day 14.23 and associated with a ten-times greater ammonium nitrate NH_4_NO_3_ input (1 L, 125 mM), gave surprising results (Figure 8). Concerning the nitrate NO_3_^−^ ion, a typical NH_4_NO_3_ fertilization increase (i.e., τ_fertilization_ ≈ 1 h, see before) was again evidenced, with a higher voltage variation and concentration multiplication ratio:
pNO_3_-ISFET: voltage variation: +30 mV, [NO_3_^−^] multiplication ratio: ≈3.5

**Figure 8 sensors-24-07143-f008:**
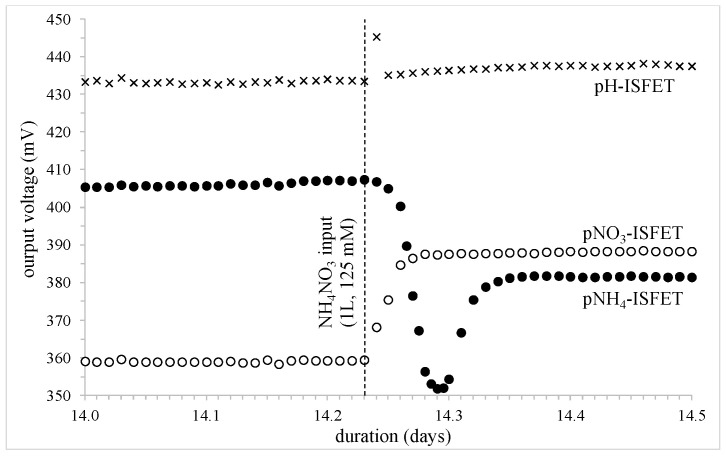
Potentiometric measurements for the different FET-based sensors (depth: −15 cm); the influence of an NH_4_NO_3_ massive input (1 L, 125 mmol).

In parallel, starting from ~407 mV, the pNH_4_-ISFET decreased drastically, in order to reach a minimal value of ~ 352 mV, and then increased to reach a maximal value of ~ 382 mV (Figure 8). This curve should be associated to contradictory phenomena, both characterized by an hourly time constant, as previously shown (τ_fertilization_ ≈ 1 h).

The first [NH_4_^+^] concentration increase should be associated to the ammonium nitrate NH_4_NO_3_ input in the soil sample. However, for the following [NH_4_^+^] concentration decrease, no compressive explanation was clearly found. Indeed, it cannot be related to a nitrification reaction by considering its high reaction kinetics. It could be assumed that the ammonium NH_4_^+^ ions supplied were initially completely available in the soil sample, before being gradually adsorbed by the negatively-charged clay-humus colloidal complex [45]. In this case, the final [NH_4_^+^] concentration decrease should be associated with some ionic electrochemical trapping occurring in soil at high concentrations.

Finally, it should be mentioned that the pH-ISFET output voltage followed an ≈5 mV increase during the whole experiment (i.e., an ≈0.1 pH total increase). This result is significant and should be taken into account in order to understand the phenomena at work, provided soil sample buffer properties are known. This last ammonium nitrate NH_4_NO_3_ input was globally characterized as follows:
pNH_4_-ISFET: voltage variation: −25 mV, [NO_3_^−^] multiplication ratio: ≈2.8

In fact, this last ammonium nitrate NH_4_NO_3_ input was planned in order to check that the different FET-based chemical sensors were operating properly, after being buried in the earth for a fortnight. The experiment was a success, demonstrating the full potential of the ChemFET technology in the frame of soil analysis.

## 4. Conclusions

Silicon-based technologies were used to develop ChemFET-based sensors for nitrogen-cycle monitoring in soil. Thus, emphasizing the use of nonactin and tetradodecylammonium nitrate (TDDAN) ionophores, respectively, pH-ChemFET, pNH_4_-ISFET, and pNO_3_-ISFET sensors were successfully fabricated, and a multi-microsensor platform was realized in order to monitor nitrogen-based ionic species in soil. Using a specific test bench, the first experiments were done in the lab, while mimicking meteorological events associated with rainy periods as well as fertilization practices. Thus, comprehensive studies were performed in an acidic (pH ≈ 4.7) clay-silt soil, characteristic of wheat fields in the south-west of France. By monitoring soil pH, pNH_4_, and pNO_3_ parameters over a fortnight (i.e., more than 4000 measurements for each FET-based sensor), different processes were successfully understood(i.e., inputs of ammonium nitrate NH_4_NO_3_) at different concentrations and water dilutions to reach soil saturation. According to experimental results, the NH_4_NO_3_ fertilization kinetics was characterized by an hourly time constant (τ_fertilization_ ≈ 1 h), whereas other phenomena were evidenced for the first time(i.e., soil nitrification (τ_nitrification_ ≈ 10 h) as well as ammonium NH_4_^+^ ion trapping in the clay-humus complex).

Such realizations and results demonstrate the potential of the ChemFET technology in the frame of the “in situ” approach for modern farming. Research has to be continued in order to further improve the measurement system, especially by studying the influence of the burying procedure on the soil(electrolyte)/sensor electrical contact. Thus, it will be possible to develop a fully-functional soil analysis (micro)system to succeed in analyzing soil nitrogen flows at different depths, have a better understanding of the different soil nitrogen-based processes, cope with nitrogen-cycle monitoring in real fields, and especially for improving nitrogen inputs in the frame of wheat culture.

## Figures and Tables

**Figure 1 sensors-24-07143-f001:**
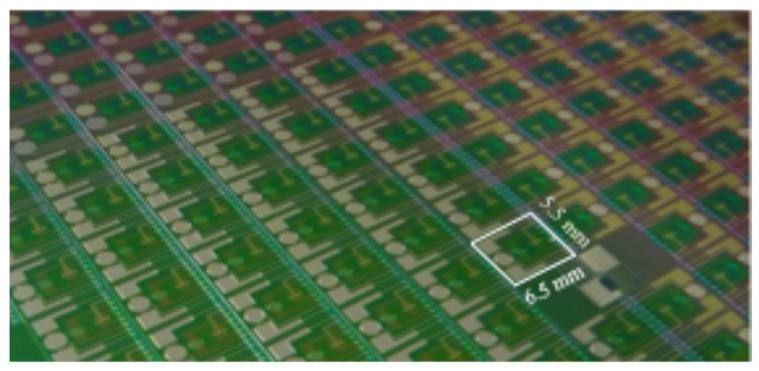
Development of silicon-based technologies for the mass fabrication of soil-analysis integrated microdevices, including a MOSFET temperature sensor, an ISFET sensor, and a microelectrode-based conductivity sensor.

**Figure 2 sensors-24-07143-f002:**
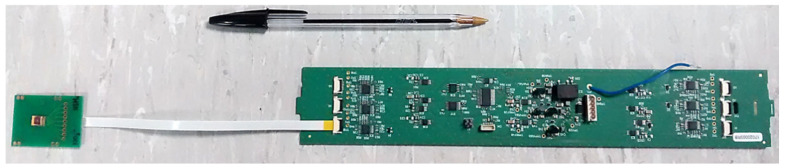
Realization of an electronic interface adapted for the measurement of six ISFET sensors.

**Figure 3 sensors-24-07143-f003:**
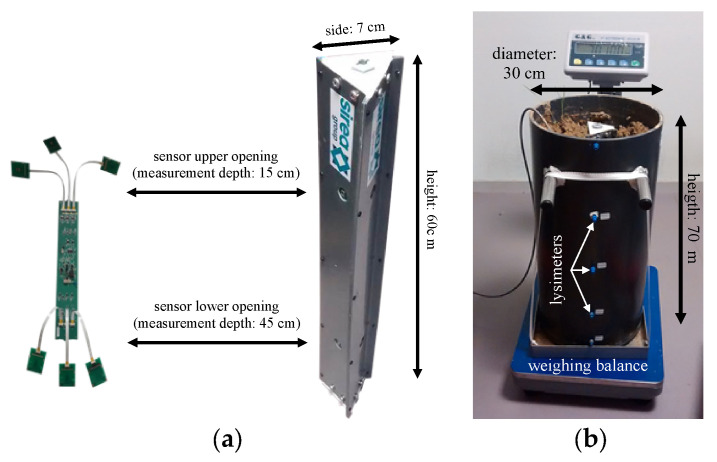
(**a**) Realization of a multi-sensor platform for the measurement of pH, pNH_4_, and pNO_3_ parameters in soils; (**b**) Test bench for the soil analysis in the lab.

**Figure 4 sensors-24-07143-f004:**
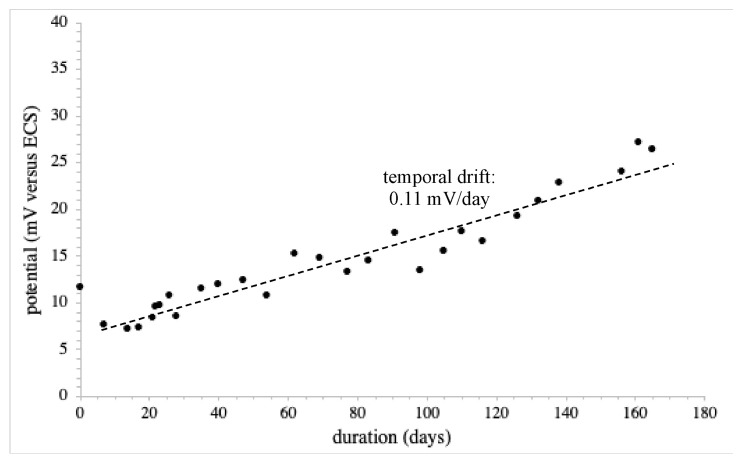
Temporal drift of the WE200 reference electrode in the acidic (pH ≈ 4.7) clay-silt soil matrix.

**Figure 5 sensors-24-07143-f005:**
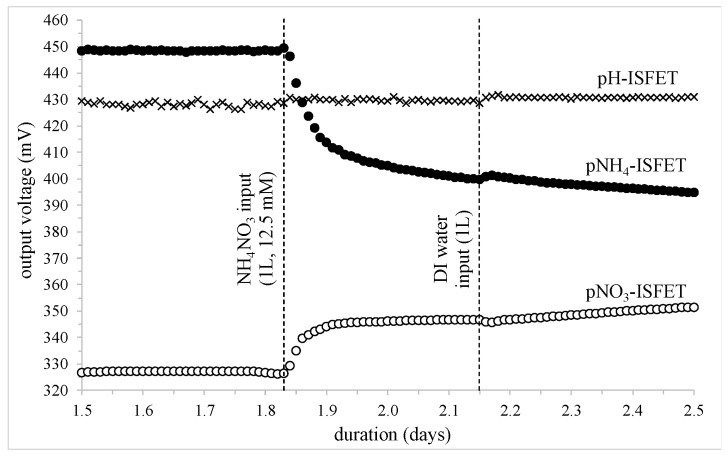
Potentiometric measurements of the different FET-based sensors (depth: −15 cm); the influence of an NH_4_NO_3_ input (1 L, 12.5 mmol) followed by a deionized water input (1 L).

## Data Availability

The original contributions presented in this study are included in the article. Further inquiries can be directed to the corresponding author.

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
