# Peer review of "Monitoring of Ammonium and Nitrate Ions in Soil Using Ion-Sensitive Potentiometric Microsensors"

_sensors, 2024, doi:10.3390/s24227143_

Round 1

Reviewer 1 Report

Comments and Suggestions for Authors

Matthieu Joly and co-authors present the use of potentiometric sensors for monitoring ammonium and nitrate ions in soil. While the sensor platform has potential, the manuscript lacks necessary clarifications and quantitative analysis. Below are specific points for improvement:

  1. Figures: Figure 1 lacks a scale bar, making it hard to interpret the dimensions of the microdevices. Consider adding the scale or moving the figure to either the main body or the Supplementary Information (SI), rather than the Materials and Methods section.
  2. Electrochemical Characterization (Section 2.4): The sensitivity of the ChemFET sensors is mentioned but not clearly defined. How was sensitivity calculated, and what method was used to quantify it? The authors should provide a more detailed explanation of the calibration process, and the exact formulae used for sensitivity derivation.
  3. Reference Electrode (Section 3.1): The characterization of the WE2000 electrode, a commercial product, does not add value to the Results section. Since its properties are known and standard, this section could be shortened or moved to the Materials section, or Supplementary information as it is not central to the experimental innovation presented.
  4. pH-ChemFET Characterization in Soil (Section 3.2): The authors claim that the pH-ChemFET sensors produced "coherent and stable" output voltages over 15 days, but this statement lacks specificity. What voltages values were considered stable, and where can this data be seen in the results? A clearer, more quantitative representation of these signals would make the findings more comprehensible and convincing.
  5. Monitoring Nitrogen-Related Ions (Section 3.3): The results for the different soil depths are not well explained. For the "-45 cm" depth, the authors report that the sensors functioned "inconsistently" or "discontinuously." It would be helpful to define what is meant by "inconsistent" and over how many days this inconsistency occurred.

Additionally, the mathematical relations provided for the voltage variations and concentration ratios (e.g., -56 mV for pNH4-ISFET and +22 mV for pNO3-ISFET) require further explanation. From which assumptions or previous work are these models derived? Including references or more details will enhance the paper's scientific rigor.

  1. Decimal Convention: The decimal format "0,06 ± 0.02 pH" should adhere to international conventions, which recommend using . for decimals  (e.g., "0.06 ± 0.02").
  2. Figures 8 and 9: Figure 8 appears to show minimal changes (approximately 10 mV) in sensor outputs during a "dry weather" period. The significance of these variations is unclear. Could the authors elaborate on the relevance of this figure and these findings? Moreover, Figure 9 is not referenced in the text near line 366, where it should be mentioned for better continuity between the results and the discussion.

By addressing these points, the manuscript will become clearer, more quantitative, and scientifically sound.

Author Response

Comment 1: Figure 1 lacks a scale bar, making it hard to interpret the dimensions of the microdevices. Consider adding the scale or moving the figure to either the main body or the Supplementary Information (SI), rather than the Materials and Methods section.

Response 1: Scales was added in the revised figure in order to precise the silicon chip dimensions.

Comment 2: Electrochemical Characterization (Section 2.4): The sensitivity of the ChemFET sensors is mentioned but not clearly defined. How was sensitivity calculated, and what method was used to quantify it? The authors should provide a more detailed explanation of the calibration process, and the exact formulae used for sensitivity derivation.

Response 2: The method for characterizing the different FET-based sensors was described in previous works (see reference [30]). That is why we do not present it in the current manuscript. We give some furthers informations in the revised version, keeping in mind the previous reference (according to fabrication reproducibility around 85%, it seems difficult to present again similar results and analytical responses).

Comment 3: Reference Electrode (Section 3.1): The characterization of the WE2000 electrode, a commercial product, does not add value to the Results section. Since its properties are known and standard, this section could be shortened or moved to the Materials section, or Supplementary information as it is not central to the experimental innovation presented.

Response 3: this part was moved in the Materials section in the corrected manuscript.

Comment 4. pH-ChemFET Characterization in Soil (Section 3.2): The authors claim that the pH-ChemFET sensors produced "coherent and stable" output voltages over 15 days, but this statement lacks specificity. What voltages values were considered stable, and where can this data be seen in the results? A clearer, more quantitative representation of these signals would make the findings more comprehensible and convincing.

Response 4: For the sake of scientific honesty, we chose to mention the poor results obtained for the ISFET sensors at the "-45 cm" depth. As a result, it was difficult to present the pH-ChemFET results at the same depth. Nevertheless, part for some minor drifts and instabilities both pH ChemFET were characterized by stable measurements, as it can be seen for the "-15cm" depth on figures 5 to 8. This was further detailed and explained in the revised manuscript.

Comment 5: Monitoring Nitrogen-Related Ions (Section 3.3): The results for the different soil depths are not well explained. For the "-45 cm" depth, the authors report that the sensors functioned "inconsistently" or "discontinuously." It would be helpful to define what is meant by "inconsistent" and over how many days this inconsistency occurred.

Response 5: As said previously, we mentioned the "-45 cm depth" experiments to show that all is not perfect for the ChemFET technology in the frame of soil analysis. Nevertheless, this part was further detailed and explained in the revised manuscript.

Comment 6: Decimal Convention: The decimal format "0,06 ± 0.02 pH" should adhere to international conventions, which recommend using . for decimals  (e.g., "0.06 ± 0.02").

Response 6: this typo (and others, sorry for that...) were corrected in the revised manuscript.

Comment 7: Figures 8 and 9: Figure 8 appears to show minimal changes (approximately 10 mV) in sensor outputs during a "dry weather" period. The significance of these variations is unclear. Could the authors elaborate on the relevance of this figure and these findings? Moreover, Figure 9 is not referenced in the text near line 366, where it should be mentioned for better continuity between the results and the discussion.

Response 7: Results associated to figure 8 (7 in the revised manuscript) were puzzling even for us. Nevertheless, since the ISFET variations were significant according to the pH-ChemFET stability and to measurement accuracy, they should be representative of some chemical phenomena in soils. After some investigations, it was finally linked to the soil nitrification processes, as explained in the manuscript. Again, this was further detailed and explained in the revised manuscript.

Finally, we would like to thank the reviewer for its comments, that help us to improve greatly our manuscript.

Sincerely.

Pierre Temple-Boyer

Reviewer 2 Report

Comments and Suggestions for Authors

Matthieu Joly etel developed Silicon based sensor to detect nitrate and ammonia from water. The manuscript is well written and informative. However, before publication, the reviewer suggests to include sensing mechanism and accuracy of the results at various temperature for the range of 5-40 oC. 

1. The state-of-the-art in the introduction section is missing. Additionally, what is the advantage of the developed ISFET over other sensors for the studied analytes?            

2. Could the authors explain the relevancy of steps shown in the experimental protocol on page 4?  

3. The shift in potential value as a function of depth is significant. Could it be explained in reference to previous literature?   

4. Can the authors provide a calibration plot of their measurement of pH, NO3, and NH4 at various simulated concentrations from low to high magnitudes? It is not clear whether the sensors will respond linearly to varying pH, NH4, and NO3 concentrations.  

5. In Figure 6, please, describe in text why NO3 increased and NH4 decreased after 1.8 days suddenly after addition of NH4NO3. Particularly, the voltage corresponding to NH4 remained stable up to 1.8 days; then, why did it decrease after the addition of NH4NO3?  

6. In Line 252, how can the authors consider the order of the curve as "first order" without demonstrating their logarithmic relation?  

7. In Lines 281-286, elaborate the claims and provide relevant references

Author Response

Comment 1: The state-of-the-art in the introduction section is missing. Additionally, what is the advantage of the developed ISFET over other sensors for the studied analytes ?

Response 1: It seems to us that the state-of-the-art was described in the Introduction section. nevertheless, it was further detailed in order to show that "microelectronics-deviated, integrated, solid-state sensors, such as ISE and ChemFET, should be developed for soil analysis. This was further detailed in the revised manuscript.

Comment 2. Could the authors explain the relevancy of steps shown in the experimental protocol on page 4 ?

Response 2: This part was further explained in the revised manuscript.

Comment 3. The shift in potential value as a function of depth is significant. Could it be explained in reference to previous literature?

Response 3: In the frame of FET-based devices, this potential shift is well-known: it is related to the threshold voltage discrepancy associated to numerous parameters of the Metal-Insulator-Semiconductor (MIS) or Electrolyte-Insulator-Semiconductor (EIS) interface. This was further explained in the revised manuscript..

Comment 4: Can the authors provide a calibration plot of their measurement of pH, NO3, and NH4 at various simulated concentrations from low to high magnitudes? It is not clear whether the sensors will respond linearly to varying pH, NH4, and NO3 concentrations.

Response 4: The method for characterizing the different FET-based sensors was described in previous works (see reference [30]). That is why we do not present it in the current manuscript. We give some furthers informations in the revised version, keeping in mind the previous reference (according to it, it seems difficult to present again similar results).

Comment 5: In Figure 6, please, describe in text why NO3 increased and NH4 decreased after 1.8 days suddenly after addition of NH4NO3. Particularly, the voltage corresponding to NH4 remained stable up to 1.8 days; then, why did it decrease after the addition of NH4NO3?

Response 5: The fact that ISFET output voltage variations are depending on the ionic valence was not clearly explained in the previous manuscript. As a result, experimental results were hardly understandable. This was further detailed in the revised manuscript.

Comment 6. In Line 252, how can the authors consider the order of the curve as "first order" without demonstrating their logarithmic relation ?

Response 6: this "first order" term was unclear and therefore unadapted. As a matter of fact, these explanations were improved in the revised manuscript by presenting the modeling of the experimental curves according to an “exp (-t/tau)" mathematical model.

Comment 7: In Lines 281-286, elaborate the claims and provide relevant references.

Response 7: For the sake of scientific honesty, we mentioned the two significant pH increases evidenced at days 1.83 and 2.14 in order to discuss them. Indeed, the "day = 1.83" seems illogical while taking into account the acidic property of the NH4+/NH3 couple (pKa = 9.23). To our knowledge, there is no relevant reference for an illogical result but his part was further explained in the revised manuscript.